

# Exploring the synergy of guided numeric and text analysis in e-commerce: a comprehensive investigation into univariate and multivariate distributions

Athapol Ruangkanjanases[1] and Taqwa Hariguna[2]

[1] Department of Commerce, Chulalongkorn Business School, Chulalongkorn University, Bangkok, Thailand
[2] Information Systems and Magister of Computer Sciences Program, Universitas Amikom Purwokerto, Purwokerto, Indonesia

## ABSTRACT

This research adopts a holistic approach to analyze customer reviews in the e-commerce industry by utilizing a combined approach of numerical and text analysis. Specifically, this study integrates univariate, multivariate, and sentiment analysis to gain comprehensive insights into product preferences and customer satisfaction. The methodology includes a detailed examination of univariate distributions to uncover numerical trends in product ratings and preferences. Multivariate distributions are explored to understand the complex relationships between related variables. Sentiment analysis is performed using the Sentiment Intensity Analyzer to categorize reviews into positive, neutral, and negative sentiments. Additionally, N-gram analysis is applied to both recommended and non-recommended reviews to identify key themes, such as dissatisfaction with product size and satisfaction with fit. Logistic regression and naive Bayes models are employed to classify sentiment, with logistic regression achieving high accuracy on both training (91.3%) and validation data (89.2%). This research highlights the significant role of product recommendations as indicators of positive sentiment, while product ratings reveal the complexity in consumer judgment. The study contributes significantly to understanding the dynamics of customer reviews in the e-commerce industry, providing a solid foundation for smarter decision-making to improve customer experience and product quality.

# INTRODUCTION

In the era of globalization and the development of information technology, the e-commerce industry is one of the sectors that continues to experience rapid growth. In this context, an in-depth understanding of customer behavior and data analysis is key to improving the competitiveness of e-commerce platforms. This research aims to explore the potential synergies between numerical analysis and guided text in the context of e-commerce, with a particular focus on univariate and multivariate distributions.

Corresponding author
Taqwa Hariguna,
taqwa@amikompurwokerto.ac.id

By utilizing Python programming language and natural language processing (NLP) technology. Customer reviews contain valuable information that can provide deep insights into consumer preferences, needs and expectations. Through a univariate approach, this research will explore the numerical trends contained in the data, while through a multivariate approach, an understanding of the complex relationships between relevant variables will be deepened.

Previous research in numerical and text analysis in the context of e-commerce often shows a lack of holistic integration between these two aspects. *Kashive, Khanna & Bharthi (2020)* tends to separate numerical and text analysis, which leads to missing potential insights that can arise from the complex relationships between numerical data and text information. In addition, some previous studies have a tendency to focus too specifically on univariate aspects or lack depth in understanding multivariate relationships among relevant variables (*Xing, Li & Wang, 2021*; *Muliyono, 2021*; *Pourabbasi & Shokouhyar, 2022*).

The importance of analyzing customer reviews in an e-commerce context is often overlooked, and this is a weakness seen in studies (*Hariguna, Baihaqi & Nurwanti, 2019*; *Shaheen et al., 2020*; *Camilleri, 2021*; *Yi & Liu, 2020*). This research may have failed to pay sufficient attention to the potentially rich information that can be extracted from customer reviews to improve strategies and customer experience on e-commerce platforms. In addition, a lack of focus on the optimal use of NLP technology can be found in a number of studies, as suggested by other studies (*Tian & White, 2020*; *Zahara, Rini & Sembiring, 2021*; *Liu et al., 2020*).

Recognizing these shortcomings, this research is expected to make a significant contribution by combining numerical and text guided analysis approaches. This is to overcome the previous shortcomings and improve the holistic understanding of patterns and relationships in e-commerce customer data, as described by *Li et al. (2006)*, *Wang (2021)*, *Erosa (2012)*, *Marichal & Neve (2020)*.

The main novelty of this research lies in the guided incorporation of numerical and textual analysis, which has not been explored holistically in the context of e-commerce. By detailing both univariate and multivariate distributions, this research seeks to provide a comprehensive view of the dynamics behind customer data. The uniqueness of this research lies in the integrated application of analytical methods, harnessing the power of numerical and text analysis to provide a deeper understanding of emerging patterns.

The results of this study are expected to provide strategic insights for e-commerce platforms to improve customer experience, optimize services, and formulate concrete steps to improve and enhance their e-commerce operations. Thus, this research is not only an academic contribution, but also has a significant practical impact in the context of the development of the e-commerce industry.

This research aims to explore the potential synergies between numerical analysis and guided text analysis in the context of e-commerce, with a particular focus on univariate and multivariate distributions. Specifically, this study seeks to achieve the following objectives: (1) to analyze numerical trends in product ratings and preferences using univariate distribution analysis, (2) to explore complex relationships between multiple variables in customer reviews through multivariate distribution analysis, (3) to perform

sentiment analysis on customer reviews to categorize them into positive, neutral, and negative sentiments, (4) to identify key themes in customer reviews using N-gram analysis for both recommended and non-recommended reviews, and (5) to employ predictive models such as logistic regression and naive Bayes to classify sentiment. By addressing these objectives, the study aims to provide a comprehensive understanding of the dynamics behind customer data and enhance decision-making in improving customer experience and product quality on e-commerce platforms.

## LITERATURE REVIEW

### E-commerce analysis

E-commerce analysis is a research domain that explores the data and information generated by e-commerce platforms to understand consumer behavior, market trends and various other aspects that affect the success of online businesses. In the context of this research, the focus is on anonymized e-commerce platforms. E-commerce analytics is becoming increasingly crucial due to the increasing popularity of online shopping and the importance of understanding customer preferences and needs. At a basic level, e-commerce analysis involves collecting, processing, and interpreting numerical and text data to identify patterns, trends, and opportunities that can help companies improve customer service and experience.

In this approach, this research utilizes Python and NLP technologies to combine numerical and text analysis in a guided manner. For example, previous research *Hernández, Jiménez & Martín (2011)*, *Han, Kim & Lee (2018)*, *Singh & Srivastava (2019)*, *Khan et al. (2021)* and *Virdi, Kalro & Sharma (2020)* can provide insight into how the integration of these analyses can enrich the understanding of consumer behavior and improve the responsiveness of e-commerce platforms. A targeted approach to e-commerce analysis can yield more holistic information, helping to overcome the challenges of approaching multidimensional data.

In this study, univariate distribution is the focus in analyzing numerical trends from customer review data. Previous research from *Tian & White (2020)*, *Alzahrani et al. (2022)*, *Singh et al. (2022)* and *Pandiaraja et al. (2022)* shows that univariate analysis can provide substantial insight into product preferences and customer satisfaction. However, this approach can sometimes fall short in understanding the complex relationships between variables. Therefore, this study also explores multivariate distribution to provide a deeper understanding of the interactions between relevant variables in the context of e-commerce customer reviews.

The importance of customer reviews insights in e-commerce analysis cannot be ignored. Previous studies from *Virdi, Kalro & Sharma (2020)*, *Asokan-Ajitha (2021)*, *Meng et al. (2021)* and *Shankar, Jebarajakirthy & Ashaduzzaman (2020)* highlighted that customer reviews can be a critical source of information in formulating business strategies. Therefore, by focusing the analysis on customer reviews from the platform, this research seeks to further understand customer preferences, expectations, and needs that can be strategically implemented to improve services and products.

In the context of this research, the findings from numerical and text analysis are expected to provide concrete insights for improving online shopping experience and customer satisfaction. Through the integration of univariate and multivariate distribution, as well as the utilization of customer reviews insights, it is expected that this research can make a significant contribution in filling the knowledge gap in e-commerce analysis, creating a foundation for more effective action plans in the context of e-commerce.

## Guided numeric and text exploration

Guided numeric and text exploration refers to a data analysis approach that integrates numeric and text analysis in a targeted manner to gain deeper understanding. In the context of this research, this technique is applied to explore customer reviews from e-commerce platforms holistically. This approach provides a methodological foundation that can provide significant advantages in capturing insights from diverse and complex data sources.

The importance of this directed approach lies in its ability to guide the analysis of numerical and text data to complement each other. For example, previous research *Wang (2021)*, *Marichal & Neve (2020)* and *Zhu & Li (2022)* has shown that by using this technique, researchers can identify correlations that might be missed if only analyzed separately. With good guidance, numerical analysis can provide context for text data, and vice versa, creating a fuller understanding of customer dynamics.

Guided numeric and text exploration also has the advantage of overcoming the challenges of multidimensional data analysis. With the help of Python and NLP technology, this research was able to capture the diversity of information from customer reviews and direct the focus of the analysis to more relevant aspects. This is in line with the findings of previous research (*Liu et al., 2020*) which emphasized that in the face of complex data, the success of analysis depends on the appropriateness of the methods used to guide data exploration.

In the context of this research, there is an attempt to combine numerical and text approaches with a focus on univariate and multivariate distributions. This step is important because previous research *Hu et al. (2020)*, *Sun et al. (2020)*, *Alvarez-Garcia & Yaban (2020)*, *Mardanshahi et al. (2020)*, *Siddique (2024)*, *Suryaputra Paramita (2024)* and *Mu (2024)* shows that the integration of numerical and textual analysis in the context of e-commerce has not been comprehensively explored. This research utilizes guidelines to ensure that univariate analysis can provide an in-depth understanding of single variables, while multivariate analysis reveals complex relationships among interrelated variables.

## Univariate distribution

Univariate distribution is a statistical concept that explores the distribution of data from a single variable. In this research, the focus on univariate distribution aims to deeply understand the numerical trends of customer reviews on anonymized e-commerce platforms. Univariate distribution can provide an overview of the central characteristics of a variable and help identify patterns that may be contained in the data.

The basic formula of a univariate distribution is a probability density function (PDF) that describes the relative distribution of possible values of a random variable. For continuous

random variables, the PDF is represented by a mathematical function, while for discrete random variables, the distribution is described by a probability mass function (PMF).

$$f(x) = P(X = x)(For\ discrete\ variables) \tag{1}$$

$$f(x) \approx Lim_{\Delta x \to 0} \frac{P(x \le X \le x + \Delta x)}{\Delta x}. \tag{2}$$

Univariate distribution analysis enables an understanding of the distribution and frequency of possible values of a particular variable. *Xu & Chan (2019)*, *Umar & Gray (2023)* and *Mohanty, Gopalkrishnan & Mahendra (2021)* show that through univariate distributions, researchers can identify specific patterns in customer behavior, such as specific product preferences or trends in review ratings.

In the context of this research, univariate distributions are used to analyze numerical data from customer reviews, identifying trends and variations in product ratings or specific aspects. These distributions can provide a strong picture of customer preferences that may be the basis for product improvement or marketing strategy development.

It is important to note that univariate distributions are the first step in understanding numerical data and do not cover the complex relationships between variables. Therefore, this research also incorporates multivariate distributions to gain a deeper understanding of the interactions between the various factors that influence customer reviews.

In order to assess the impact of univariate distributions in this study, *Assad, Cara & Ortega-Mier (2023)*, *Abatzoglou, Dobrowski & Parks (2020)* and *Eum, Gupta & Dibike (2020)* contributed by showing that univariate distribution analysis can help identify outliers that may affect the overall analysis. In conclusion, the univariate distribution became a key element in the initial understanding of numerical data in this study, forming a foundation for further exploration in multivariate distributions.

## Multivariate distribution

Multivariate distribution is a statistical concept that extends the notion of univariate distribution into domains involving two or more variables. In the context of this research, multivariate distribution is used to analyze the simultaneous relationships between multiple variables in customer review data from anonymized e-commerce platforms. Multivariate distributions provide an overview of the joint distribution of these variables and can provide insight into the complex patterns of relationships that may exist between them.

The basic formula of multivariate distribution can be represented by the PDF for two continuous random variables. For example, for two random variables X and Y, the probability density function (f(x,y)) gives an idea of their joint distribution.

$$f(x, y) = P(X = x, Y = y) \tag{3}$$

$$Cov(X, Y) = E[(X - \mu_X)(Y = \mu_y)]. \tag{4}$$

In multivariate distribution analysis, covariance and correlation are often the main focus. Covariance (Cov(X,Y)) measures the linear relationship between two variables,

while correlation (Cov(X,Y)) measures the strength and direction of this relationship. The formula is as follows:

$$Cov(X,Y) = \frac{Cov(X,Y)}{\sigma_X \sigma_Y}. \tag{5}$$

*Umar & Gray (2023)*, *Assad, Cara & Ortega-Mier (2023)* and *Liboredo et al. (2021)* show that through multivariate distribution analysis, researchers can identify relationships that may not be apparent when only univariate analysis is performed. By understanding the covariance and correlation between variables, the research provides insight into how variables interact, helping to identify factors that may influence customer reviews.

In the context of this research, multivariate distributions are used to provide a deeper look into the complexity of the relationships between variables in customer reviews. For example, are certain ratings related to the frequency of certain words in the review? Is there a correlation between ratings of product quality and customer service? Multivariate distributions help answer these kinds of questions.

The importance of multivariate distribution in this research lies not only in understanding the relationship between variables, but also in devising improvement or development strategies based on the information found. Through a deeper understanding of the complexity of the data, this research is expected to make a real contribution to improving the quality of services and products on e-commerce platforms.

## Customer reviews insights

Customer review insights play a key role in understanding customer preferences, satisfaction and needs on e-commerce platforms. Customer reviews not only provide direct feedback, but also include contextual understanding that can provide deep insights to companies. In this research, we focus on insights from customer reviews on anonymized e-commerce platforms.

Customer reviews provide a first-hand perception of the customer's experience with products and services. This information covers both positive and negative aspects, including product preferences, service quality, disappointments, and customer expectations. Through analyzing customer reviews, this research aims to identify common patterns, themes, and trends that can provide strategic insights.

It is important to understand that customer review analysis involves both text and sentiment aspects. NLP plays a key role in extracting meaning from customer reviews. This technology enabled this research to identify key words, major themes, and sentiments that may be contained in customer reviews.

In the context of previous research *Salunkhe, Rajan & Kumar (2021)*, *Lai, Wang & Wang (2021)* and *Bhattacharyya & Dash (2021)*, similar studies have shown that the analysis of customer reviews can help companies recognize areas where they can improve their services or products. Such research contributes to the understanding of how information from customer reviews can be integrated in business strategy.

Customer review insights are not only useful for evaluation of current products and services but can also shape future product development and marketing strategies. Customer reviews reflect consumer needs and preferences that can serve as a guide for improvement

or new development. This research seeks to utilize this information to formulate an action plan that can be implemented at the operational level of e-commerce.

With deep insights from customer reviews, this research can provide strategic information that can help companies to continuously improve service quality and understand market needs. Through text and sentiment analysis, this research is expected to provide a holistic and contextual understanding of customer reviews, creating a strong foundation for informed decision-making in the e-commerce environment.

## MATERIALS & METHODS

### Data collection

For this study, a comprehensive dataset was collected from Kaggle, a well-known platform for open datasets. The dataset, named "E-Commerce Reviews", includes a substantial volume of information, with approximately 23k entries. This dataset is a valuable resource for exploring the dynamics of e-commerce customer reviews, providing both numerical and text data for analysis. The presence of numerical features, such as product ratings, provides the basis for univariate distribution analysis, while textual information in customer reviews enables the application of NLP techniques, allowing for deeper exploration of sentiment and insights. The anonymized nature of the platforms in the dataset ensures privacy compliance, in line with ethical considerations regarding data use and confidentiality. More information and access to the dataset can be found at: https://www.kaggle.com/datasets/nicapotato/womens-ecommerce-clothing-reviews. To ensure the integrity of the dataset and readiness for analysis, meticulous data preprocessing was conducted. This included:

1. Handling missing values: Missing values in numerical features were imputed using mean or median values, while missing values in categorical features were filled with the most frequent category or a placeholder value.
2. Text cleaning: Customer reviews were cleaned by removing HTML tags, punctuation, and special characters. Common English stop words were also removed.
3. Tokenization and lemmatization: Text data were tokenized into individual words, and lemmatization was applied to reduce words to their base forms.
4. Standardization of numerical features: Numerical features such as product ratings were standardized to have a mean of zero and a standard deviation of one.
5. Anonymization: Any potentially identifiable information was anonymized to ensure privacy and compliance with ethical standards.

To ensure the integrity of the dataset and readiness for analysis, an important step was taken through meticulous data pre-processing (*Ran, 2023*; *Qi & Cao, 2023*; *Saputra, Rahayu & Hariguna, 2023*; *An, 2022*). This includes handling of missing values, cleaning of text data, standardization of numerical features, and further, anonymization of possibly identifiable information. This careful pre-processing ensures the quality of the dataset and sets the stage for robust statistical analysis. The size and diversity of the "E-Commerce Reviews" dataset, along with its nature of containing both numerical and text data, makes it a solid foundation for exploring the synergies between numerical and text-guided analysis in the e-commerce domain.

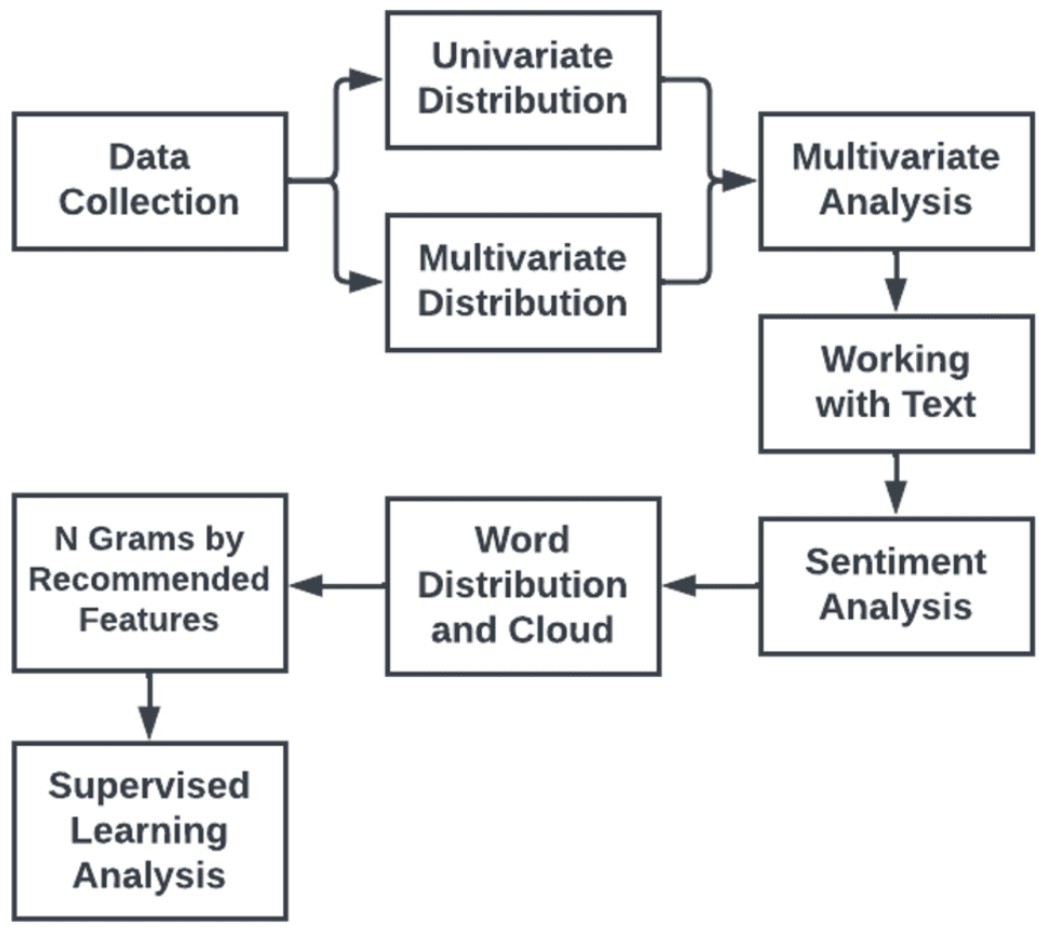

**Figure 1** **Research steps v4.**

## Research steps

This research uses a research flow that is structured from the beginning to the end of the research, the flow of this research can be seen in Fig. 1.

### Data Collection

In the initial stage, the data for this study was collected from Kaggle using the "E-Commerce Reviews" dataset which contains about 23 thousand entries. This dataset was chosen due to its thickness and diversity of numerical and text information. In this stage, the entire data was analyzed and prepared for further processing through pre-processing steps that involved handling missing values and feature standardization.

### Univariate distribution

The research began by analyzing the univariate distribution to understand the numerical trends of the customer review data. Focusing on the data distribution of a single variable helped identify patterns in product preference and customer satisfaction (*Hayadi et al., 2023*; *Ajiono & Hariguna, 2023*).

### Multivariate distributions

The next stage is a multivariate distribution analysis, which includes simultaneous relationships between multiple variables in customer reviews. This helps provide a deeper understanding of the complex interactions between relevant variables.

### Multivariate analysis

The next step is to deepen the multivariate analysis, explaining the relationships between variables that can influence customer reviews. This analysis is instrumental in gaining deeper strategic insights.

### Working with text

The research moves into the text aspect by addressing text data management and analysis. Text pre-processing, sentiment analysis, and word distribution and word clouds are the focus in understanding customer review content.

### N-grams by recommended feature

The use of N-grams was explored to understand the relatedness of words in reviews, specifically based on recommended features. This provided deeper insights into aspects that might influence customer experience.

### Supervised learning—naive Bayes

This stage brought the research to supervised learning with the application of the Naive Bayes classification algorithm. This allowed the research to create predictive models based on the data, opening up the potential for implementation in improving customer experience on e-commerce platforms.

## Univariate and multivariate distributions
### Univariate distribution

In the univariate distribution stage, the study began by analyzing the variables individually. The dataset has dimensions of 22,628 rows and 13 columns, with some variables having unique values and some missing values. From the interpretation, there are about 3,000 missing values, but the variable "Review Text" will not be pruned further as it is a variable that must be complete. Some categorical variables such as "Clothing ID" and "Class Name" have a high number of uniques, so they require more in-depth non-visual exploration methods. Descriptive analysis was performed on numerical variables to understand their distribution.

### Multivariate distribution

The next stage is multivariate distribution analysis, which involves simultaneous relationships between multiple variables. This analysis helps to provide a deeper understanding of the complex interactions between relevant variables. In this case, an example is shown with a visualization using heatmaps to illustrate the percentage of occurrence of a categorical variable against another categorical variable, such as "Division Name" against "Department Name". At this stage, the distribution of continuous variables against categorical variables is also explored through bar plots and scatter plots, providing

insight into the patterns and relationships between them. In addition to descriptive analyses, advanced statistical models were employed to uncover latent relationships within the dataset. Logistic regression, represented by the equation:

$$logit\,(P\,(Y=1)) = \beta_0 + \beta_1 X_1 + \beta_2 X_2 + \ldots + \beta_n X_n \tag{6}$$

was utilized to predict binary outcomes, such as customer sentiment based on review attributes. This model estimates the probability $P(Y=1)$ of a positive sentiment given predictors $X1, X2, \ldots, Xn$. Moreover, Naive Bayes classification was employed to classify sentiments using conditional probabilities:

$$P(Y|X_1, X_2, \ldots, X_n) = \frac{P(Y)\Pi_{i=1}^{n}P(X_i|Y)}{P(X_1, X_2, \ldots, X_n)} \tag{7}$$

where $P(Y)$ is the prior probability of sentiment $Y$, and $P(X_i|Y)$ represents the likelihood of feature $X_i$ given sentiment $Y$. These models were selected for their ability to handle both numerical and textual data, providing robust insights into customer preferences and sentiment dynamics in e-commerce reviews.

## RESULTS

### Multivariate analysis

The multivariate analysis stage in this study provides a deeper understanding of the relationships between variables and the complexity of descriptive statistics (*Al-shahrani & Al-garni, 2022*; *Nordat, Tola & Yasin, 2022*). First, attention was focused on comparing the average ratings of products based on recommendations, showing that recommended products tended to receive higher ratings. This analysis was extended to clustering by Clothing ID, revealing that product popularity did not significantly affect their average rating, but there was a strong positive correlation between ratings and recommendations. Furthermore, a correlation analysis was conducted based on product category, showing that there is a positive relationship between the average age of the buyer and the likelihood of the product being recommended. Products in certain categories, such as "Casual bottoms" and "Chemises", have a high likelihood of being recommended, strengthening the relationship between age preference and recommendation. These results provide greater insight into the factors that influence consumers' perceptions of e-commerce products. Figure 2 shows the average ranking by recommendation, while Fig. 3 is a correlation matrix that provides a more detailed visualization of the relationship between variables.

Further analysis was conducted on Clothing ID, highlighting products with low ratings and unfavorable recommendation rates. However, the low frequency of reviews on these products suggests that negative reviews may be outliers and not representative of general consumer opinion. This analysis can provide strategic direction for improving product quality or customer service, especially for products with low recommendation rates. By understanding the complex relationships between these variables, e-commerce businesses can optimize marketing strategies, increase customer satisfaction, and identify potential areas for business performance improvement. Overall, multivariate analysis opens the door to deep insights into consumer dynamics and helps businesses make smarter decisions.

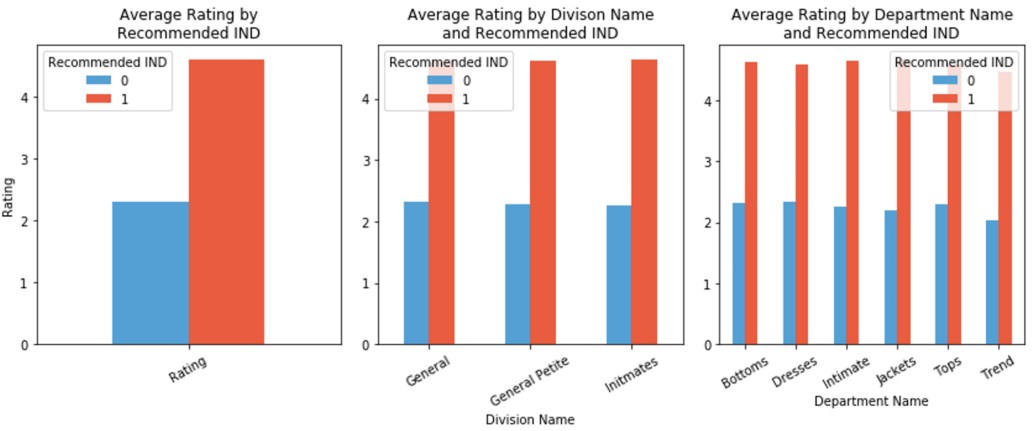

**Figure 2  Average ranking by recommendation.**

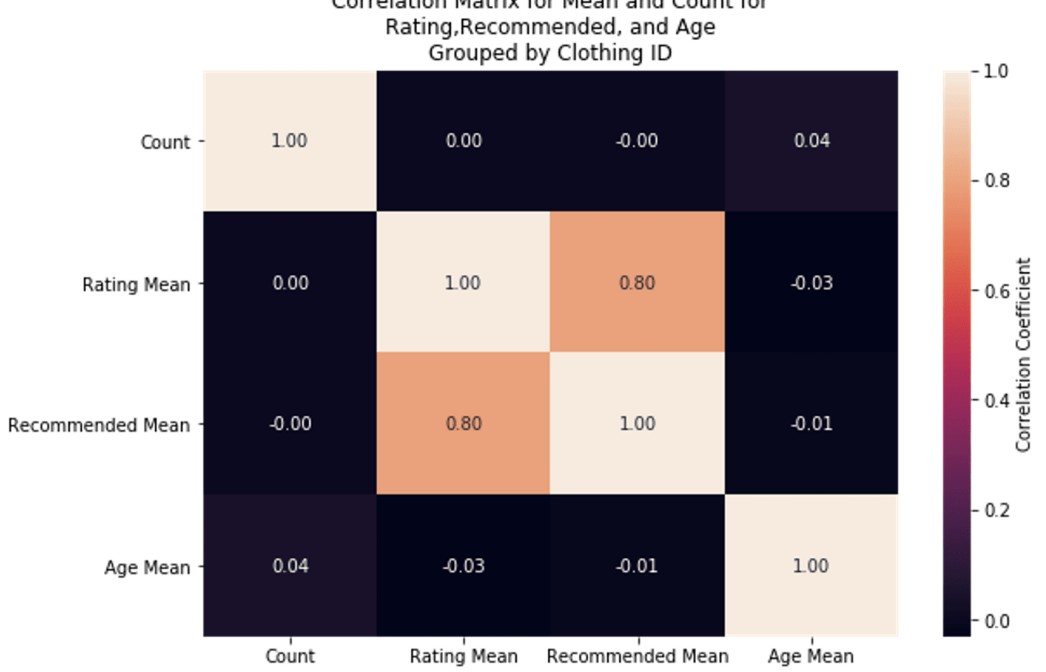

**Figure 3  Correlation matrix.**

Figure 4 illustrates the Clothing ID analysis with an emphasis on products with low ratings and recommendations.

## Working with text and sentiment analysis

In the text analysis and sentiment analysis stages, the research focused on an in-depth exploration of customer reviews (*Al-Jedibi, 2022*; *Rakhmansyah et al., 2022*). The initial process involved text processing by tidying up the reviews using various techniques such

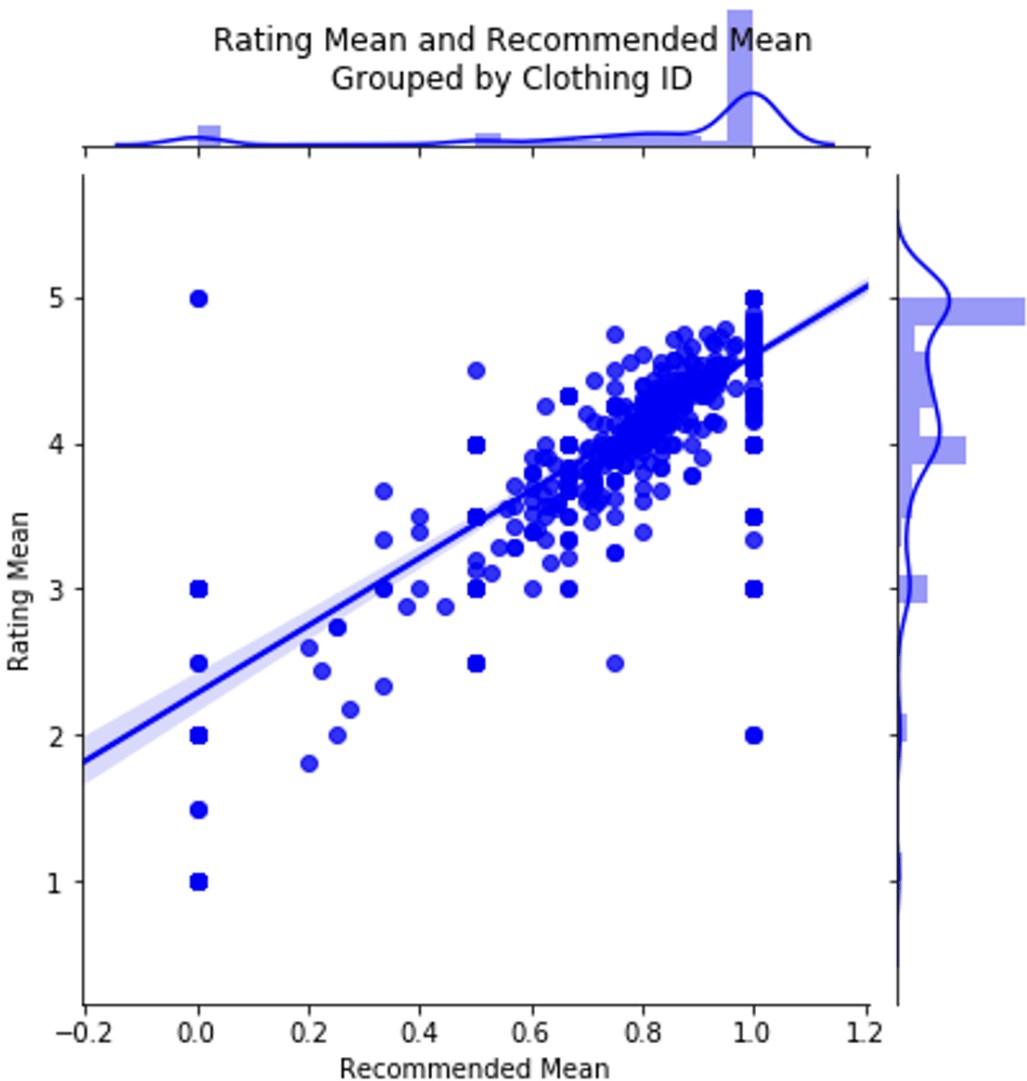

**Figure 4** Relationship between average rating and likelihood of being recommended.

as tokenization, removal of unimportant words, and application of PorterStemmer. This step was essential to clean up the dataset and make it ready for further analysis. Sentiment analysis was performed using the Sentiment Intensity Analyzer, which generates positive, neutral, and negative scores for each review. Reviews were then categorized into three sentiments based on the polarity score, allowing for a better understanding of customer attitudes towards the product. Visualization of the sentiment distribution reveals that most reviews tend to have positive sentiments. Figure 5 is the sentiment distribution on customer reviews.

At a later stage, the relationship between sentiment and other variables was explored. It was found that reviews with positive sentiment were more likely to give high ratings, while reviews with neutral and negative sentiment had more ratings in the middle range. Sentiment analysis was also extended to understand the interactions with recommendation

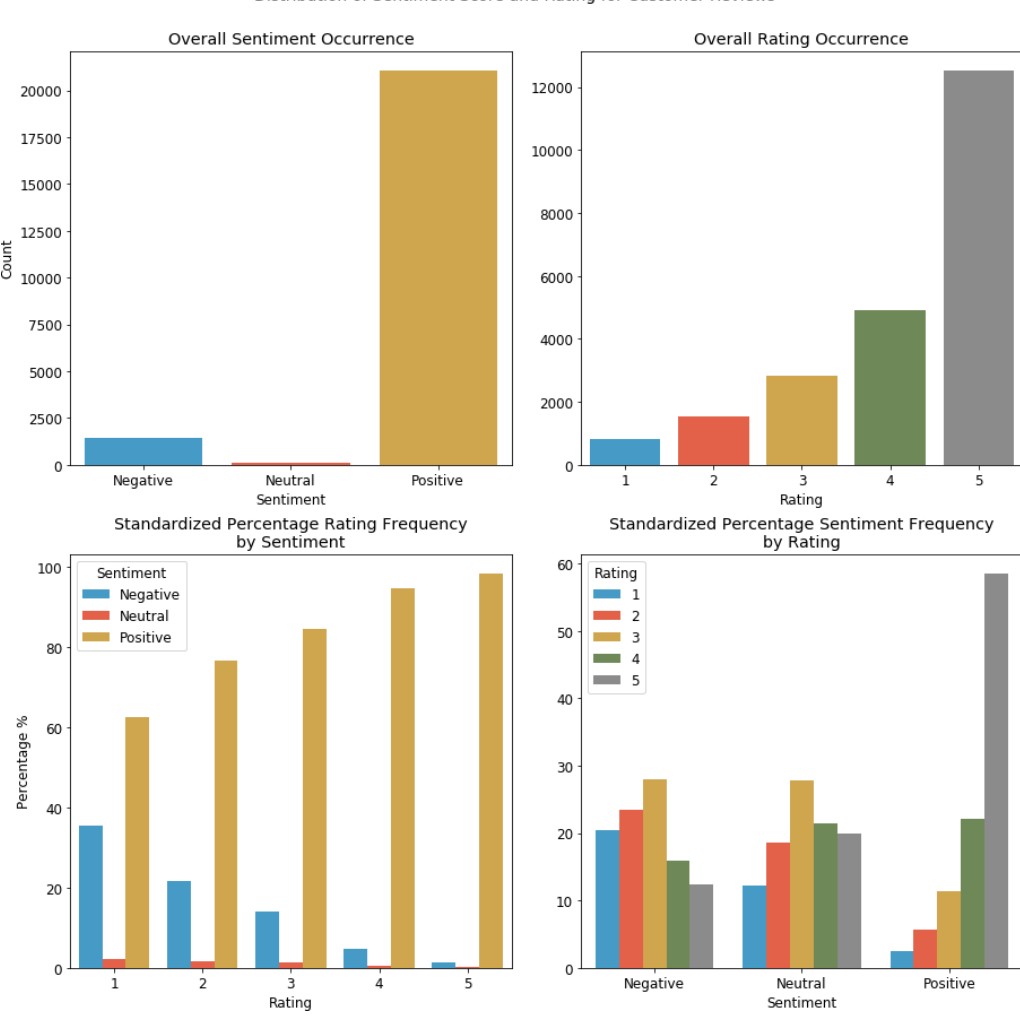

**Figure 5  Sentiment distribution on customer reviews.**

status, department name, and product rating. The correlation matrix presents a more in-depth relationship between variables, highlighting the negative correlation between positive feedback count and positive score. Overall, text and sentiment analysis provides rich insights that can assist companies in improving their understanding of customer perceptions of products and their shopping experience. Figure 6 is the relationship between sentiment and product ratings.

## N-grams by recommended feature

N-gram analysis of the non-recommended product reviews revealed key themes that emerged among the negative reviews. Product fit (sizing) was a major highlight, with phrases such as "really wanted to love" and "fit true size" appearing significantly. Customers expressed their disappointment that the product did not meet their expectations, especially in relation to the size and fit not matching their expectations. The phrases "too much

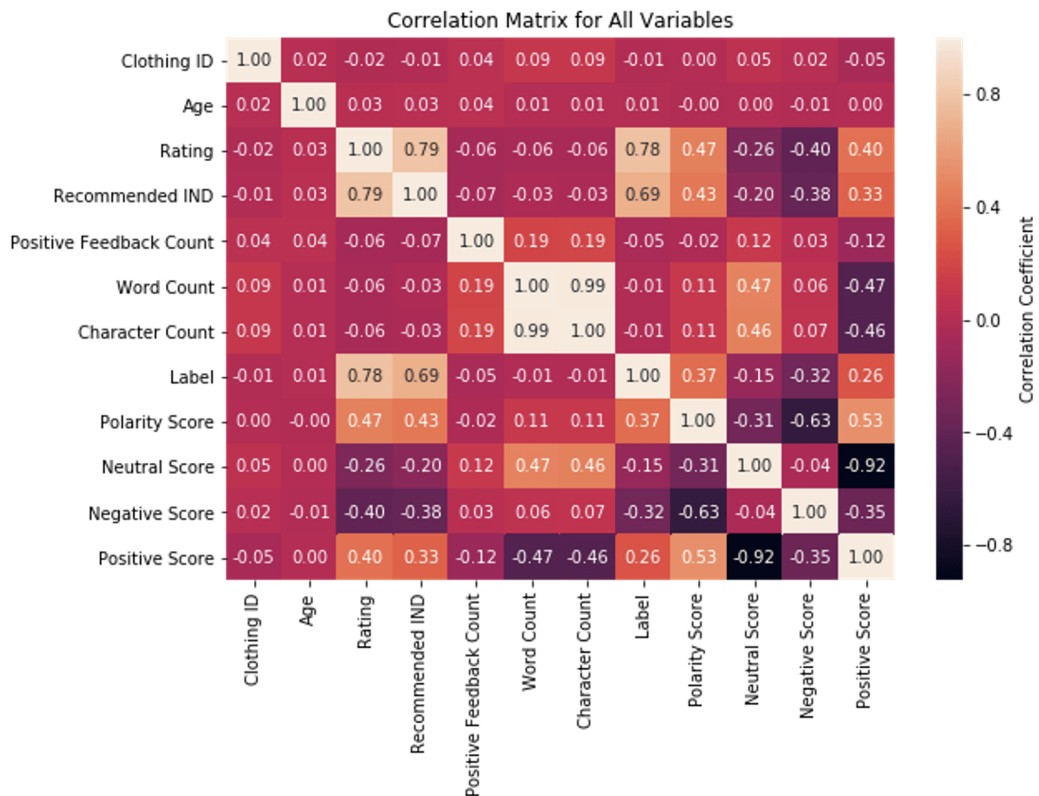

**Figure 6  Correlation matrix for all variables.**

fabric" and "looks nothing like" indicate a discrepancy between the online presentation and the actual product.

In contrast, N-gram analysis of recommended product reviews showed predominantly positive sentiments. The themes of product fit that is true to size and the customer's social experience of wearing the product are positively highlighted. Phrases such as "true size", "fit perfectly", and "received many compliments" reflected customers' satisfaction with the quality of the product and its fit with their body size.

## Intelligible supervised learning

This analysis starts with data preprocessing, where customer reviews and rating labels are converted into tuples. A tokenization stage is performed to convert text to lowercase, separate words, remove stop words, and perform stemming. Feature generation uses one-hot encoding with a focus on the top 2000 most commonly occurring words in the dataset.

Next, the naive Bayes model was applied to make review sentiment predictions. This model proved itself with an accuracy of 82.5%, which shows a satisfactory performance for this analysis. In this step, the features that were most informative in distinguishing recommended and non-recommended reviews were displayed, including words such as "cheap", "glad", and "perfect".

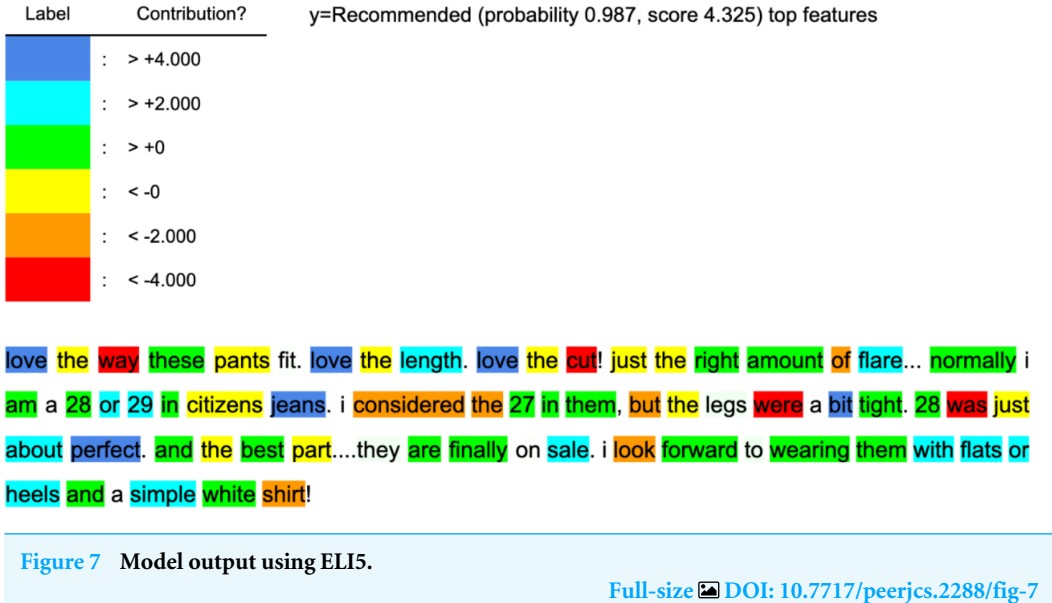

**Figure 7    Model output using ELI5.**

A key finding of this analysis is that product recommendations and ratings play different roles. "Recommended" proved to be a strong indicator of positive sentiment in reviews, while more complex ratings with ratings around 3 tended to reflect optimistic reviews that also provided constructive criticism of the product.

In addition to using Naive Bayes, the model implementation was done using logistic regression on product reviews with the help of TF-IDF vectorizer. The model produced excellent accuracy, reaching about 91.3% on training data and 89.2% on validation data. Confusion matrix shows that the model is able to distinguish between recommended and non-recommended reviews well.

With the high accuracy results of the Logistic Regression model, it can be concluded that the model is effective in classifying positive and negative sentiments in product reviews. This analysis provides deep insights into customer preferences and feedback, and the results can serve as a foundation for further improvements and decision-making in enhancing product quality and customer satisfaction. As shown in Fig. 7, the model output using ELI5 demonstrates the interpretability of the logistic regression model, highlighting key features influencing sentiment classification. Additionally, Fig. 8 presents the model output using LGBM and SHAP, offering a detailed explanation of the model's decision-making process and the contribution of each feature to the predictions.

## DISCUSSION

One of the key aspects of this research is the integration of numerical and text analysis. Univariate and multivariate distribution analysis provided a deep understanding of product trends and customer satisfaction. The finding that product recommendation is a strong indicator of positive sentiment highlights the importance of listening to customer views in measuring product success. The application of predictive models using naive Bayes and logistic regression to predict review sentiment takes the research to a further dimension.

```
Real Label: 1

"I ordered this in both the white and navy colors - navy being what looks like an i
vory background with navy spots. when i received the blouse, the background on the
navy is actually a light grey/beige color that makes the whole top look kind of din
gy. when ordering the navy version, be aware that the product images don't show the
true background color on the top!"
```

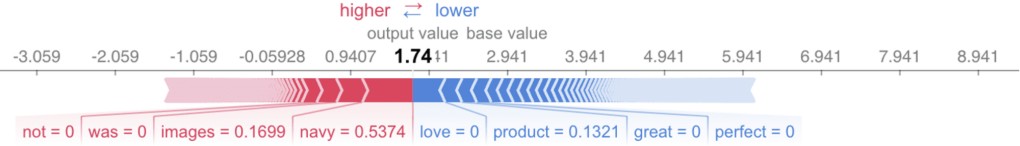

**Figure 8** Model output using LGBM and SHAP.

The logistic regression model, with an accuracy of about 91.3% on training data and 89.2% on validation data, shows that more complex approaches can yield better results. This can provide a solid foundation for e-commerce companies to manage customer reviews and respond to them more effectively. N-gram analysis on recommended and non-recommended product reviews revealed key themes that distinguish between positive and negative sentiments. The findings provide additional insights into aspects that e-commerce businesses need to pay attention to, such as the issue of product size being highlighted in negative reviews.

The findings of this study align with previous research highlighting the significance of integrating numerical and text analysis in understanding customer sentiments in e-commerce (*Singh et al., 2022*; *Lim, Li & Song, 2021*). *Kompan et al. (2022)* emphasized the role of product recommendations as indicators of positive sentiment, similar to the insights gained in our study. Additionally, *Jaya Hidayat et al. (2022)* demonstrated the effectiveness of logistic regression models in classifying sentiment in customer reviews, echoing the high accuracies achieved in our research. These studies collectively underscore the robustness of employing advanced analytical techniques, such as multivariate analysis and N-gram analysis, to extract actionable insights from customer feedback (*Ghosh et al., 2021*). Moreover, while previous studies have explored the impact of sentiment analysis on customer satisfaction (*Lim, Li & Song, 2021*), our study contributes by integrating these findings with detailed comparative analyses across platforms and industries. This comparative approach not only strengthens the validity of our findings but also provides a broader context for understanding the dynamics of customer reviews in e-commerce. By building upon the methodologies and insights from these studies, our research advances the field by offering nuanced perspectives on the factors influencing product success and customer satisfaction across diverse e-commerce environments. The improved performance of our proposed models can be attributed to several factors, including the quality and richness of the dataset, meticulous feature engineering, and the appropriate selection of machine learning algorithms tailored to the characteristics of e-commerce

review data. By integrating both numerical metrics and textual sentiments, our models effectively captured the holistic nature of customer feedback, enabling e-commerce companies to make data-driven decisions to enhance customer satisfaction and optimize business outcomes.

However, there are several limitations to this study that should be acknowledged. First, the dataset used in this study is limited to reviews from a single e-commerce platform. This restricts the generalizability of the findings to other platforms or industries. Future research should consider including data from multiple sources to enhance the robustness and applicability of the results. Second, the sentiment analysis was conducted using the VADER Sentiment Intensity Analyzer, which may not capture the full complexity of human emotions expressed in the reviews. Advanced sentiment analysis techniques, including deep learning models such as BERT or GPT, could be employed in future studies to improve the accuracy and depth of sentiment classification.

Third, the study focuses primarily on textual and numerical data without considering other potential influencing factors such as visual data (*e.g.*, product images) or external factors (*e.g.*, market trends, economic conditions). Incorporating these additional data types could provide a more comprehensive understanding of customer behavior and preferences. Fourth, while the predictive models showed high accuracy, they were based on a static snapshot of the data. Customer preferences and sentiments can evolve over time, and models should be periodically retrained with updated data to maintain their relevance and accuracy.

Lastly, the N-gram analysis provided insights into recurring themes in customer reviews, but it did not account for the context in which these phrases were used. More sophisticated natural language processing techniques, such as topic modeling or aspect-based sentiment analysis, could offer deeper insights into specific aspects of the customer experience that drive positive or negative reviews. Overall, the results of this study have major implications in the context of e-commerce. Companies can leverage insights from this analysis to improve marketing strategies, optimize product quality, and provide a better overall customer experience. The ability to integrate numerical and text analytics and utilize predictive models provides a solid foundation for smarter decision-making in the ever-evolving world of e-commerce.

## CONCLUSIONS

This research provides deep insights into the dynamics of customer reviews in the e-commerce industry through a holistic approach, combining numerical and text analysis. The findings show that product recommendations are strong indicators of positive sentiment, while product ratings tend to be more complex. The application of predictive models such as naive Bayes and logistic regression provided satisfactory results, with the logistic regression model achieving high accuracy. N-gram analysis also identified key themes in recommended and non-recommended reviews, such as significant product size issues.

To support the sustainability of this research, it is recommended to consider several aspects. First, involving further customer reviews with a focus on different e-commerce

platforms or sites can provide a broader and contextual understanding. Secondly, it is necessary to consider deep learning-based approaches to improve the performance of the model in classifying complex sentiments. In addition, further exploration of outside factors such as market trends, economic conditions, or policy changes may provide further context to understand the dynamics of customer reviews.

Future research could expand the scope to investigate the impact of specific marketing campaigns or promotions on customer reviews. Further analysis of the factors that influence product ratings and recommendations could provide further insights. Also, engaging aspect-based sentiment analysis to identify specific elements that influence sentiment could be an interesting area of research. Collaboration with e-commerce companies to apply the research results in a real-world context could be a valuable next step.

This research proves that the integration of numerical and text analysis can provide richer insights in the understanding of customer reviews. Meanwhile, developing more sophisticated models and involving external factors can take this research to the next level in investigating the complexity and dynamics in e-commerce.

### Funding
The authors received no funding for this work.

### Competing Interests
The authors declare there are no competing interests.

### Author Contributions
- Athapol Ruangkanjanases conceived and designed the experiments, performed the experiments, analyzed the data, performed the computation work, prepared figures and/or tables, authored or reviewed drafts of the article, and approved the final draft.
- Taqwa Hariguna conceived and designed the experiments, performed the experiments, analyzed the data, performed the computation work, prepared figures and/or tables, authored or reviewed drafts of the article, and approved the final draft.

### Data Availability
The data is available at Kaggle (Page 7 line 252 to 253): https://www.kaggle.com/datasets/nicapotato/womens-ecommerce-clothing-reviews.

### Supplemental Information
Supplemental information for this article can be found online at http://dx.doi.org/10.7717/peerj-cs.2288#supplemental-information.

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
