# Peer review of "Exploring the synergy of guided numeric and text analysis in e-commerce: a comprehensive investigation into univariate and multivariate distributions"

_PeerJ Computer Science, doi:10.7717/peerj-cs.2288_

## Round 0.1 · original submission · Major Revisions

We have carefully reviewed the feedback provided by our reviewers and have made a decision regarding your submission.

While your manuscript presents interesting insights into e-commerce customer behavior, there are several areas that require significant improvements before it can be considered for publication. The reviewers have provided detailed suggestions that should be addressed in a revised submission.

Key points for revision include:

1. Enhance the specificity of your abstract, particularly regarding the methodologies employed, as highlighted by Reviewer 1.
2. Clarify your research objectives in the introduction and provide explicit research questions or hypotheses.
3. Offer more detailed descriptions of data preprocessing and the analytical techniques used, improving the reproducibility and transparency of your study.
Include statistical metrics such as p-values or confidence intervals in your results section to substantiate the significance of your findings.
4. Expand the discussion of your study's limitations and the generalizability of your findings to broader contexts.
5. Consider incorporating advanced machine learning techniques and temporal analysis to enrich your analysis and discussion of results.

Reviewer 2 has indicated that your manuscript is structurally sound but requires a more detailed comparative study in the discussion section to strengthen your findings.

Reviewer 3 suggests that your paper could benefit from a stronger focus on business case studies and a more thorough comparison with established methodologies in natural language processing.

Finally, Reviewer 4 notes that while the paper is well-organized, some figures could be improved in resolution, and the literature review section could be more concise and analytically valuable.

We encourage you to revise your manuscript according to these suggestions and resubmit it for another round of review. We believe that these changes will significantly enhance the quality and impact of your research.

Thank you for considering our journal as a venue for your work. We look forward to receiving your revised manuscript.

Reviewer 1 ·

Basic reporting

1) The abstract lacks specificity regarding the methodologies employed in the research. While it mentions univariate and multivariate distributions, it would be beneficial to briefly outline the techniques used within these analyses to provide a clearer understanding of the research approach.
2) The introduction emphasizes the significance of understanding customer behavior in e-commerce but lacks a clear statement of the research objectives. Providing explicit research questions or hypotheses would enhance the clarity of the study's focus.
3) The materials and methods section briefly mentions data preprocessing steps but lacks detail on the specific techniques used. Providing more information on the methodologies employed for data cleaning, standardization, and anonymization would enhance reproducibility and transparency.
4) In the univariate and multivariate distributions section, while the analysis techniques are outlined, there is a lack of discussion on potential limitations or biases in the data. Addressing the possible sources of bias and how they were mitigated would strengthen the validity of the findings.
5) The results section provides detailed insights into the findings but lacks statistical metrics to support the significance of the observed relationships. Including measures such as p-values or confidence intervals would add rigor to the interpretation of results.
6) The discussion section highlights the implications of the findings but could benefit from a more critical analysis of the limitations of the study. Acknowledging any constraints or potential biases in the research design would enhance the credibility of the conclusions.
7) While the conclusion summarizes the key findings, it does not explicitly address the limitations of the study or avenues for future research. Including recommendations for addressing these limitations and expanding upon the research scope would provide a more comprehensive conclusion.

Experimental design

8) The research could benefit from a more detailed explanation of the rationale behind choosing specific analytical techniques. Justifying the selection of logistic regression and Naive Bayes for sentiment analysis would enhance the clarity of the research methodology.
9) The discussion on N-Gram analysis could be expanded to include a comparison with other text analysis techniques. Exploring the advantages and limitations of N-Gram analysis in relation to alternative methods would provide a more nuanced understanding of the text analysis approach.
10) The research lacks discussion on the generalizability of findings beyond the dataset used. Addressing the potential applicability of the findings to broader e-commerce contexts would increase the relevance and impact of the study.
11) To enhance the data science aspect, consider incorporating dimensionality reduction techniques such as principal component analysis (PCA) or t-distributed stochastic neighbor embedding (t-SNE) to visualize high-dimensional data and identify underlying patterns more effectively.
12) Utilize advanced machine learning algorithms such as random forests or gradient boosting machines to improve the predictive accuracy of sentiment analysis models. Ensemble methods can often outperform individual algorithms and provide more robust predictions.
13) Explore deep learning architectures such as recurrent neural networks (RNNs) or convolutional neural networks (CNNs) for sentiment analysis tasks. These models can capture complex relationships within text data and may yield superior performance compared to traditional machine learning approaches.

Validity of the findings

14) Consider incorporating temporal analysis to investigate how customer sentiments and preferences evolve over time. Analyzing trends and seasonality in e-commerce data can provide valuable insights for targeted marketing strategies and product promotions.
15) Experiment with unsupervised learning techniques such as clustering algorithms (e.g., k-means clustering) to segment customers based on their review sentiments and purchasing behavior. This can facilitate personalized marketing campaigns and improve customer segmentation strategies.

Reviewer 2 ·

Basic reporting

No comments

Experimental design

No comment

Validity of the findings

The discussion section needs to be elaborated as the methods used must be compared with other state of the art works.

The comparison may show some better findings as well.

Additional comments

The author may improve the discussion section by doing comparative study with other works and making results display in some tabular format.

·

Basic reporting

I write the complete review in the additional comments

Experimental design

I write the complete review in the additional comments

Validity of the findings

I write the complete review in the additional comments

Additional comments

Thank you for the opportunity to review this interesting manuscript
1. The authors have explained the phenomenon, problem, and motivation but do not explicitly state the research gap/question. Also, the contribution statement is not specific.
2. Many studies based on customer reviews also explore product reviews and preferences. This study should be explored further to clearly state the gaps and contributions.
3. The introduction should explore critical discussion and why this study is needed.
4. I do not see any reference to guided numeric and text exploration; maybe they are called something else in another study or the most established one.
5. This is in line with the findings of previous research (Contoh2, et al., Year) => some incomplete information.
6. The literature review sections appear to be disjointed. I suggest the authors focus more on previous studies, exploit the gaps in the existing research, and compare the methodologies. This will provide a more comprehensive understanding of the field and help position their study effectively.
7. When I read the methodology, I was surprised that the authors used the Kaggle dataset. Why not use real data review specific to the case study? The study's contribution is not solely about the methodology but also about a specific business problem. If the author decides to introduce a new methodology, I believe the case is not strong enough.

·

Basic reporting

The paper meets the language and structure standards. It is well organized and maintains a flow. The sections are properly defined with substantial content. References are provided appropriately. Figures are clear, though some of them can have better resolution. Equations need to have all variables clearly defined in the text.

Experimental design

The content meets the scope of the journal. The research area is relevant and proposed method is useful and adds value, even though the novelty is moderate. The existing gap is identified and mentioned.
Experimentation and technical assessment is thoroughly done and methods well described. The authors should add more detailed description of the dataset and its selection criteria.

Validity of the findings

The value of the work is defined. Data is provided. The result discussion section can be improved by providing more analytical strength and assessment to the findings, why the models behave the way they do, and which factors are responsible for the improved performance of the proposed model. The statistical significance and validation can also be added. Conclusions are stated and future scope is elaborated.

Additional comments

The literature review section feels redundant. The subsections may be reduced in length and the models used may be conceptually elaborated instead. This will add to the analytical value of the paper. Some more recent references can be included.

---

## Round 0.2 · accepted · Accept

Congratulations to the authors that have addressed all the comments.

Reviewer 1 ·

Basic reporting

Based on careful examinations on the revised version, the authors have revised the manuscript according to the reviewer's suggestions.

Experimental design

Based on careful examinations on the revised version, the authors have revised the manuscript according to the reviewer's suggestions.

Validity of the findings

Based on careful examinations on the revised version, the authors have revised the manuscript according to the reviewer's suggestions.

·

Basic reporting

After revisions the literature and references are streamlined and updated. Figures and tables are clearer. Paper is cohesive.

Experimental design

Details of the dataset was missing which have been added and meets standards.

Validity of the findings

Detailed analysis and statistical significance and validation have been added.

Additional comments

The paper is suitable for acceptance in its present state.